# Prediction at Ungauged Catchments through Parameter Optimization and Uncertainty Estimation to Quantify the Regional Water Balance of the Ethiopian Rift Valley Lake Basin

Tesfalem Abraham [1,2,*], Yan Liu [2,3], Sirak Tekleab [1] and Andreas Hartmann [2,3]

[1] Department of Water Resources and Irrigation Engineering, Institute of Technology, Hawassa University, Hawassa P.O. Box 05, Ethiopia

[2] Faculty of Environment and Natural Resources, University of Freiburg, 79098 Freiburg, Germany

[3] Institute of Groundwater Management, Dresden University of Technology, 01069 Dresden, Germany

* Correspondence: tesfalem.abraham@hydrology.uni-freiburg.de

**Abstract:** Quantifying uncertainties in water resource prediction in data-scarce regions is essential for resource development. We use globally available datasets of precipitation and potential evapotranspiration for the regionalization of model parameters in the data-scarce regions of Ethiopia. A regional model was developed based on 14 gauged catchments. Three possible parameter sets were tested for regionalization: (1) the best calibration parameters, (2) the best validation parameter set derived from behavioral parameters during the validation period, and (3) the stable parameter sets. Weighted multiple linear regression was applied by assigning more weight to identifiable parameters, using a novel leave-one-out cross-validation technique for evaluation and uncertainty quantification. The regionalized parameter sets were applied to the remaining 35 ungauged catchments in the Ethiopian Rift Valley Lake Basin (RVLB) to provide regional water balance estimations. The monthly calibration of the gauged catchments resulted in Nash Sutcliffe Efficiencies (NSE) ranging from 0.53 to 0.86. The regionalization approach provides acceptable regional model performances with a median NSE of 0.63. The results showed that, other than the commonly used best-calibrated parameters, the stable parameter sets provide the most robust estimates of regionalized parameters. As this approach is model-independent and the input data used are available globally, it can be applied to any other data-scarce region.

**Keywords:** data-scarce region; parameter estimation; uncertainties; ungauged catchment; weighted regression; water balance

## 1. Introduction

Global freshwater is particularly stressed by rapidly growing human populations and all the negative consequences of environmental change. Hydrological quantification of freshwater resources is crucial to managing and mitigating these impacts and thus promoting benefits from these resources. As such, there is a growing need for the accurate monitoring and simulation of water balance components to support and maximize water resource management practices. In Ethiopia, most of the freshwater lakes are located in the Great East African Rift Valley, which was formed by volcanic depressions and cracks during the Pliocene [1]. The region is known for its scarce and limited hydro-climatic data, which have limited regional understanding of water balance processes. In many developing regions of the world, hydro-climatic stations are not sufficiently established due to limited economic and technological development [2]. In the Ethiopian Rift Valley Lake Basin (RVLB), even the data from the few available gauging networks are of poor quality, contain gaps, and are subjected to human disturbances. Consequently, the region has remained one of the least studied in Ethiopia. Due to data scarcity, there is a lack of hydrologic

simulations using available rainfall-runoff models in poorly managed catchments around the world. New approaches that use global datasets and quantify the uncertainties of regionalization would therefore be helpful for these data-scarce regions.

The Prediction for Ungauged Basins (PUB) initiative aims in particular at developing strategies for better understanding and reduced uncertainty in data-scarce regions [3,4]. Studies have started to focus on ungauged catchment predictions during the PUB decade [4,5] and in the post-PUB decade [6], including those comparing existing regionalization approaches for large sample studies [7]. Two general approaches have been used for predictions in ungauged basins: the first estimates model parameters from calibrated model parameters based on selected objective functions [8–11]; the second is a model-independent approach, which uses streamflow signatures to establish constraints that can describe the physical and climatic characteristics of watersheds [12]. The latter has been shown to reduce uncertainties that can emerge from the model structural error [13–17]. This approach has shown skill in predicting the expected streamflow in ungauged catchments, especially when going along with the quantification of uncertainties emerging from observed streamflow data [18]. However, recent work also showed that the information content of streamflow signatures is limited [19]. A recent study also analyzed the calibration of the relationship between model parameters and catchment properties, rather than regionalizing the best-calibrated parameter [20]. So far, however, most of these approaches have only been applied in data-rich regions of the world. The credibility and validity of such strategies have yet to be well tested in poorly recorded climatic data regions such as ours.

Rainfall-runoff models are used to represent the typical physical and climatic properties of a catchment. The physical representation of the available rainfall-runoff models may range from parsimonious–spatially lumped to complex physical–spatially distributed models. The common problem with most rainfall-runoff models is that they require some sort of parameter estimation to provide robust predictions. Most model parameters are not directly measurable or linkable to the physical properties of the given catchment because of model simplifications or disagreements between the model scale and the observation scale (incommensurability) [21–23]. However, an inherent correlation between the model parameters and catchment properties can often be assumed [5,7,24,25]. Model parameters represent the characteristics of the complex catchment system that are difficult to measure on a small scale. The accurate representation of catchment properties by model parameters should be evaluated, to some degree, by the selected objective function, which measures the fit between observed discharge and simulated discharge. However, most of the catchments are ungauged and their parameter estimations will be subject to uncertainties. Despite the aforementioned limitations in data-scarce regions, there are some studies that have developed strategies to derive model parameters at the global scale using various regionalization approaches [26–28]. In addition, different hydrological models have been employed for the regionalization studies [6]. Yet, the obtained simulations often show a lack of precision due to errors emerging from global input data quality and regionalization uncertainty. Uncertainties would be particularly propagated due to the regionalization method itself and the human interference in the catchments [28]. Li and Zhang [29] have also found huge uncertainty from the grid parameter regionalization for regions with rare gauging sites. Another approach estimates regionalized parameter sets on a global scale by using data from catchments distributed mainly in temperate regions [30]; their implementation and validity in the tropics remain questionable because their regionalization procedure does not include many catchments from this climatic region.

In this study, the applicability of high-resolution global climate data was tested for deriving parameters of the ungauged catchments through regionalization and demonstrating a novel leave-one-out spatial cross-validation procedure to quantify regionalization uncertainties for the region in the RVLB with scarce precipitation data. To reduce the regionalization uncertainties, the weighted least square regression was applied that accounts for the errors introduced by unidentifiable model parameters [31]. This brings more advantages

to the regression model by increasing the representation of catchments with identifiable parameters. Other than the typical approach of using the best-calibrated parameters of the gauged catchments, e.g., Wagener and Wheater [10]; Singh et al. [32]; Lane et al. [33], the idea of using multiple similar parameter sets for regionalization was extended. Previous studies already considered multiple similar parameter sets for regionalization [34] and calibrated the relationship between the model parameters and catchment properties [20]. However, to our knowledge, the differences between using the best-calibrated parameter, the best parameter set in the validation period, and the most stable parameter set considering their performance in the calibration and validation period have not yet been explored. A spatial split-sample test was used to evaluate the best-calibrated parameter, the best parameter set in the validation period, and the most stable parameter set considering their performance in the calibration and validation period for their adequateness for regionalization. We presented this new approach to quantify uncertainty in regional model development that can be adapted for data-scared regions. We have also demonstrated the applicability of global climate datasets for deriving the regional model. Rather than using a common best-calibrated parameter set to derive the regional model, we compared three different approaches to find the best regional parameter sets. Although our approach uses a low number of catchments, we demonstrate acceptable regional simulation performance. Thus, the objective of this study is to identify the most reliable regional model that can be derived from the three different best parameter sets. In addition, the study hypothesized that a parameter set that is stable over time (based on its performance during calibration and validation) could lead to a more stable spatial extrapolation of the parameters among the three approaches for deriving the best parameter sets. Due to the limited number of observations of streamflow data, this approach is implemented to a relatively low number of 14 gauged catchments with reliable streamflow data to estimate the water balances of 35 ungauged catchments in the RVLB. Repeating the leave-one-out spatial split-sample test multiple times, the uncertainty was quantified that goes along when regionalizing parameter sets from a low number of catchments. That way, this study provides useful directions for regional modeling and uncertainty quantification in under-represented and data-scarce regions such as the RVLB, where assessments of the impacts of climate variability and climate changes are most urgently needed. The remainder of the paper is outlined as follows: A description of the study area and the datasets used in Section 2; A model description and the overall parameter estimation and evaluation procedure in Section 3; A comparison of regionalization results in Section 4; Discussion in Section 5; The summary of conclusions in Section 6.

## 2. The Study Area

The RVLB is located in the southern part of the Main Ethiopian Rift Valley and covers an area of 53,035 km$^2$, providing water supply to a population of more than 15 million people who live mainly on subsistence agriculture (Figure 1). The RVLB is 84 km wide, and adjacent to it there are large, discontinuous Miocene-aged normal faults [35,36]. Within the Main Ethiopian Rift, there are a series of right-stepping, quaternary rift basins, which have faulted magmatic segments, extending about 20 km wide and 60 km long, that are embryonic oceanic spreading centers. The central part of the RVLB is formed by a Pliocene-aged, faulted caldera, caused by a fractured volcano [1]. Existing faults and repeated ground cracks at the floor of the caldera have increased the permeability of the rock. Within the basin, several small-to-medium-sized catchments drain into eight freshwater lakes. For most of these catchments, there is a lack of hydro-climatic data; what data are available contain gaps and are subject to human interventions. In recent decades, the RVLB has experienced major droughts and extreme flooding due to rainfall variability, making prediction difficult [37]. This has resulted in an uncertain analysis of high and low flows, factors that are important for quantifying the hydrological water balance components in this region.

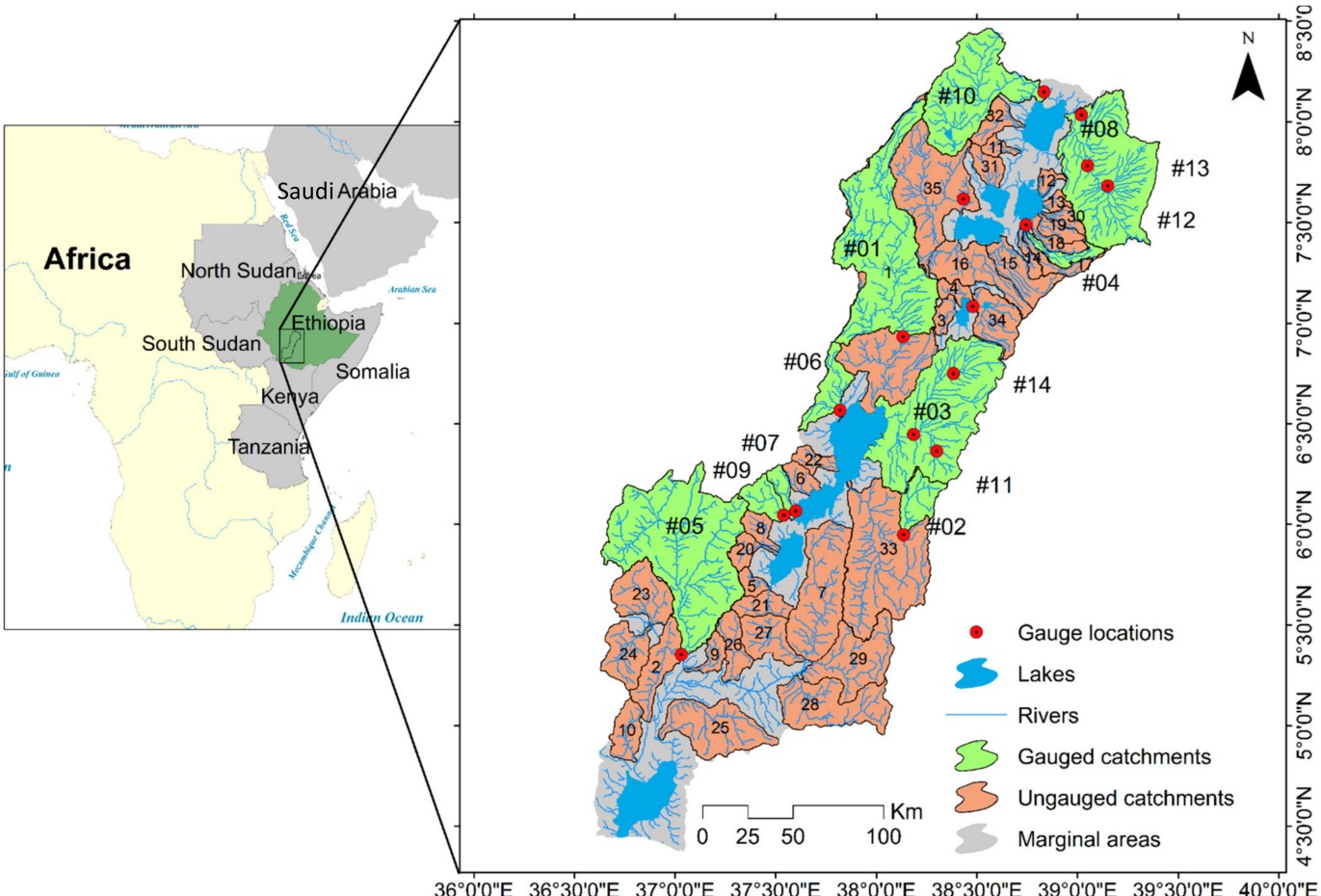

**Figure 1.** The study basin showing 14 gauged catchments for the regional model development and 35 ungauged catchments that are draining to the respective lakes through the river networks.

The climate is characterized by the range from semi-humid to semi-arid subtropical climate in the central and southern part of RVLB with a mean annual temperature of 20 °C [38]. In the same region, the long-term analysis of annual average precipitation shows a variation in the range of 951 mm to 1653 mm, and such variations are attributed to the topographic differences in the region [38]. However, catchments in the northern part of RVLB are characterized as sub-humid and humid climates, with average annual precipitation of 1050 mm and temperatures of around 15 °C [39,40].

*Data and Catchment Properties*

The regionalization procedure used precipitation products of Multi-Source Weighted-Ensemble Precipitation (MSWEP) version 2 and evapotranspiration from the Global Land Evaporation Amsterdam Model (GLEAM V3) (Table 1). MSWEP products have recently developed precipitation datasets at a finer scale (0.1°), which are constructed using different sets of precipitation data from gauges, monthly satellite data, and re-analysis data on the global scale. The global MSWEP data have been evaluated and showed better performance for data-scarce regions including Ethiopia [41]. The data product has been widely applied for the evaluation of gridded satellite products and regionalization studies in data-scarce regions [42]. For the discharge data due to monitoring and organization issues, high-quality streamflow data are only available from 1995 to 2007. The data for the recent period (2008–2015) are mostly not available for the user, and the available parts are of poor quality. We, therefore, collect the daily streamflow data for the period from 1995 to 2007 from the Ministry of Water Irrigation and Energy of Ethiopia (MOWIE) for 14 catchments in the

RVLB. The 14 catchments provide a sufficient length of data (>10 years) for these simulation periods. The area of 14 catchments ranges from 144.2 to 4528.2 km².

**Table 1.** Variables showing the climatic and physiographic data and their resolutions and periods.

| Variable | Spatial Resolution | Time Period | Temporal Resolution | Source | Reference |
|---|---|---|---|---|---|
| Precipitation | 0.1° | 1995–2007 | Daily | MSWEP V2 | Beck et al. [41] |
| Potential evapotranspiration | 0.25° | 1995–2007 | Daily | GLEAM v3 | Martens et al. [43] |
| HBV-parameters | 0.5° | - | - | www.gloh2o.org (access date 12 January 2020) | Beck et al. [30] |
| Elevation | 30 m | - | - | SRMT V2.1 | https://earthexplorer.usgs.gov (access date 28 January 2020) |
| Wetness index (P/PE) | Point scale | 1995–2007 | Daily | MSWEP V2 and GLEAM v3 | Beck et al. [41]; Martens et al. [43] |
| Streamflow | Pont scale | 1995–2007 | Daily | MOWIE | - |

Catchment properties are common descriptors of the hydrological process and are frequently applied to estimate model parameters in ungauged catchments. However, there is no general rule to select suitable properties. The available option should be a selection of as many catchment descriptors as possible while reducing correlated catchment properties so as to exclude redundant information and obtain independent variables. In this regard, the main criteria to select catchment properties is that they are model-independent and can be used for model predictions to be hydrologically realistic in both gauged and ungauged catchments [12]. For reliable regionalization, a sufficient number of catchment properties should be selected. Their selection, in turn, depends on data availability, hydrologic relevance, and the suitability of the properties. With these considerations, nine catchment properties were derived from the physical and/or climatic information in both gauged and ungauged catchments from the RVLB. Due to the lack of local land cover and soil property data, the catchment properties that need to be derived from those data are not included in the analysis. Physical catchment properties such as drainage area, drainage density, mean catchment slope, mean elevation, and catchment index were extracted from the Digital Elevation Model (DEM). Climatic properties of potential evapotranspiration and precipitation that usually affect the rainfall-runoff process were extracted from the global data sets (Table 2 and Table S2). Other physical properties where local information is not available, such as permeability and porosity, were extracted from the global datasets prepared by Huscroft et al. [44]. These dominant catchment properties are also widely applied in the previous regionalization studies using the HBV model structure [7,45,46].

Each parameter in the HBV model has a certain physical interpretation that can be linked with the selected catchment properties. Previous study has also shown that a sufficiently large correlation coefficient between catchment property and model parameter can be a good indicator of the predictive power of the selected catchment properties, provided there is no collinearity between them [47]. Considering this, nine catchment properties were carefully selected that will be linked with the physical interpretation of the calibrated HBV model parameters (see Section 3.1). By undertaking this, catchment properties that have redundant information and may not be suitable for regionalization were also removed.

**Table 2.** Descriptions and values of properties for gauged catchments in the RVLB that were used for the development of the regional model.

| Cat No and Names at the Gauge Location | Catchment Properties | Drainage Area [km²] | Drainage Density [km km⁻²] | Mean Slope [%] | Mean Elevation [m] | Catchment Index [m km⁻¹] | Permeability [log₁₀ m²] | Porosity [-] | Wi [-] | P [mm] |
|---|---|---|---|---|---|---|---|---|---|---|
| | Description of properties | Index of catchment area | The ratio of catchment stream length to the drainage area | Mean of the percentage slope for each terrain unit | Index describing the mean of catchment elevation | Mean of all inter-nodal slopes in a catchment | Index describing the nature of water flow in the shallow aquifer | The fraction of the volume of voids in the shallow aquifer | Wetness index (Wi) as the ratio of precipitation (P) to potential evapo-transpiration (PE) | Annual average precipitation (1995–2007) |
| #01-Bilate@Tena | | 3821.2 | 0.075 | 16.22 | 2037.1 | 10.07 | −12.194 | 0.07 | 0.85 | 923.6 |
| #02-Gelana@Tore bridge | | 506.4 | 0.124 | 24.17 | 2084.5 | 10.39 | −12.5 | 0.09 | 1.17 | 1309.1 |
| #03-Gidabo@Measso | | 2590 | 0.113 | 20 | 1805.4 | 14.54 | −12.248 | 0.097 | 0.85 | 942.07 |
| #04-Gedemso@Langano | | 241.5 | 0.67 | 18.1 | 2759.3 | 28.11 | −12.5 | 0.09 | 0.88 | 919.19 |
| #05-Woito@Bridge | | 4528.2 | 0.07 | 29.09 | 1439.5 | 10.64 | −11.433 | 0.028 | 1.34 | 1319.6 |
| #06-Hamassa@Wajifo | | 534.4 | 0.34 | 15.86 | 1655.5 | 15.22 | −12.306 | 0.076 | 0.94 | 1208 |
| #07-Hare | | 196.5 | 0.36 | 33.08 | 2343.1 | 77.67 | −12.2 | 0.076 | 0.897 | 1107.5 |
| #08-Katar@Abura | | 3241.1 | 0.115 | 8.7 | 2601.9 | 19.99 | −12.034 | 0.064 | 0.69 | 779.8 |
| #9-Kulfo@Arbaminch | | 397.2 | 0.226 | 36.39 | 2249.9 | 76.55 | −12.283 | 0.08 | 1.52 | 1617.8 |
| #10-Meki@Meki village | | 2033.1 | 0.111 | 19.27 | 2124.4 | 11.12 | −12.155 | 0.068 | 0.58 | 667.2 |
| #11-Gidabo@Bedesa | | 144.2 | 0.341 | 30.18 | 2149.7 | 56.74 | −12.5 | 0.09 | 1.29 | 1397.7 |
| #12-Katar@Fete | | 1940.9 | 0.117 | 14.49 | 2668.9 | 17.78 | −12.171 | 0.075 | 0.87 | 991.2 |
| #13-Katar@Timela | | 205.2 | 0.65 | 18.29 | 2953.5 | 40.87 | −12.371 | 0.084 | 0.7 | 785.6 |
| #14-Gidabo@Aposto | | 491.8 | 0.327 | 21.49 | 2012.6 | 26.76 | −12.483 | 0.089 | 1.18 | 1300.1 |

## 3. Methods

The concept of the regionalization procedure is shown in Figure 2. Global climatic forcings of precipitation and potential evapotranspiration were applied for parameter estimation in the gauged catchments using the HBV hydrologic model. The precipitation products of the Multi-Source Weighted-Ensemble Precipitation (MSWEP) version 2 and evapotranspiration of the Global Land Evaporation Amsterdam Model (GLEAM V3) were used for the study. Parameter sets that perform best during calibration and validation and ones that are stable between calibration and validation (stable parameter sets) were derived. Using the best parameter sets from calibration, a correlation analysis was conducted with the physical and climatic properties of gauged catchments, which forms the basis for the regression. By applying the best parameters from the calibration, validation, and stable sets, three regression models were derived in the leave-one-out procedure and the one that performed best was selected. The parameters of the ungauged catchment were estimated by applying the ungauged catchment properties to the regression models (Figure 2). In addition, the regionalization uncertainty was quantified through the spatial cross-validation procedure that applies the leave-one-out method (see Section 3.4). Consequently, the parameters derived from the regression were evaluated by comparing the discharge observations and simulations of the gauged catchments on their calibration (1995–2002) and validation periods (2003–2007). Comparing their performances during the calibration and validation period enables the selection of the best regional models derived from the parameters estimated from calibration, validation, and stable sets.

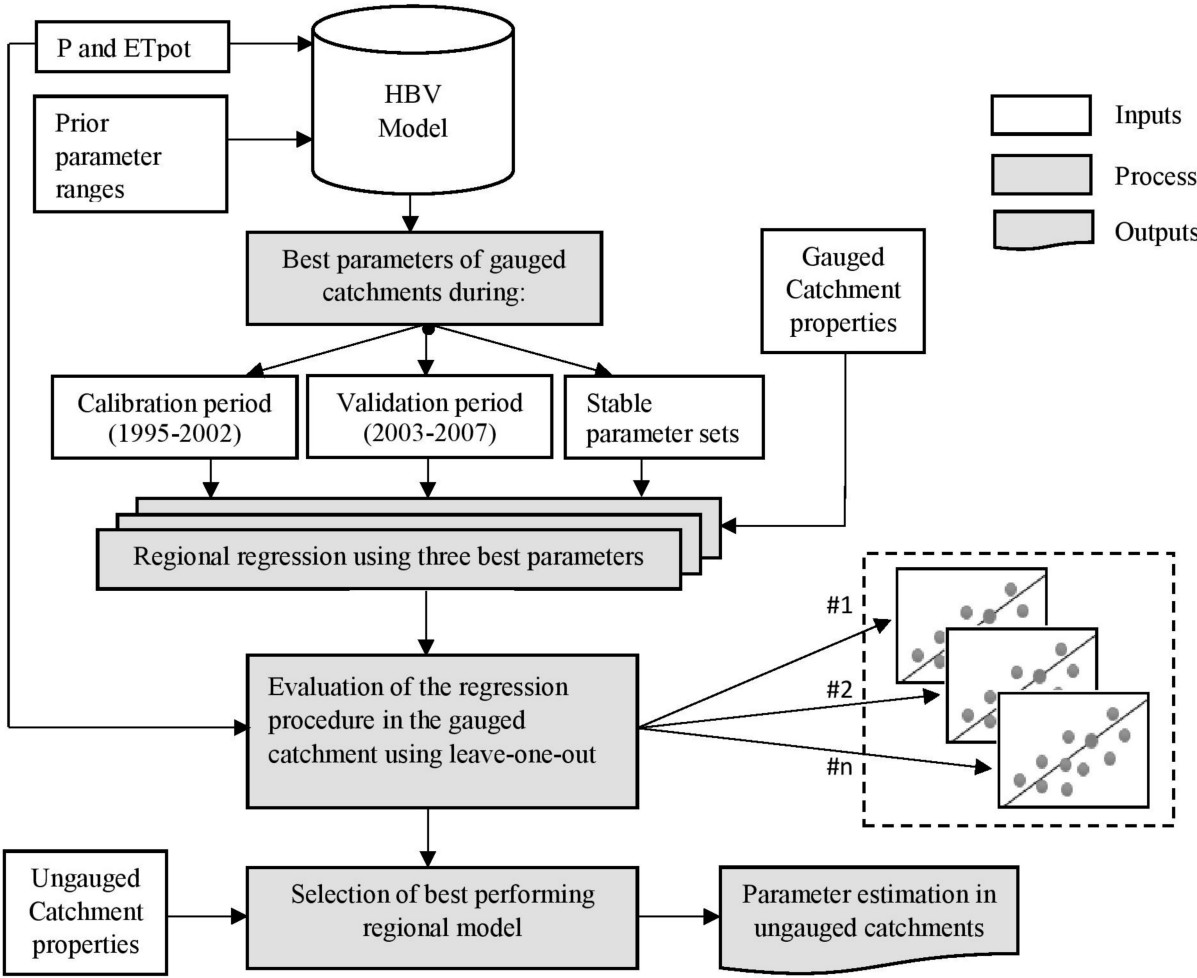

**Figure 2.** Schematic diagram showing the entire procedure applied in this study.

*3.1. Hydrological Model*

In this study, a lumped HBV hydrologic model was used [48,49], which has been applied in a wide range of climatic and physiographic conditions [50,51]. The HBV model has been tested in various parts of the world and frequently applied in several regionalization studies due to the simplicity and flexibility of its model structure [24,25,45,52–55].

This study applied the HBV model modified by Zhang and Lindström [51] and Beck et al. [30]. The model was run at a daily time scale using daily inputs of precipitation and potential evapotranspiration. The lumped HBV model uses effective/average precipitation in the catchment domain as input. Since the aim of this study is to explore the possibility of using global precipitation products in data-scare regions, the gridded MSWEP V2 precipitation data were used. The data have been evaluated with separate gauge-radar data and have undergone bias correction so that the influence of elevation was indirectly considered while producing the gridded precipitation data. Due to the absence of snow processes in the RVLB, the model consisted of routines of soil moisture accounting, runoff response, and a channel routing procedure, which are controlled by nine model parameters (Table 3). Three calibrated parameters, $\beta$, $F_C$, and $L_P$, control the soil moisture dynamics. $\beta$ controls the contribution ($dQ$) to the runoff response routing and the increase ($dP\text{-}dQ$) in soil moisture storage ($Ssm$) and $F_C$ is the maximum soil moisture storage in the model as shown by Equation (1). $L_P$ is the value of the soil moisture above which evapotranspiration ($Ea$) reaches its potential level ($Ep$). The actual evaporation from the soil moisture zone equals the potential evaporation if $Ssm/F_C$ is above $L_P * F_C$ as shown by Equation (2).

$$\frac{dQ}{dP} = \left(\frac{S_{sm}}{F_C}\right)^{\beta} \tag{1}$$

where $dP$ and $dQ$ are precipitation and runoff [mm d$^{-1}$].

$$E_a = E_P . min\left(\frac{Ssm}{F_C . L_P}, 1\right) \tag{2}$$

**Table 3.** HBV parameter ranges for the RVLB and their descriptions, derived from Beck et al. [30].

| Parameter | Description | Global Range (Min to Max) |
|---|---|---|
| $\beta$ [-] | Shape coefficient of recharge function | 1–6 |
| $F_C$ [mm] | Maximum water storage in unsaturated-zone store | 50–700 |
| $K_0$ [d$^{-1}$] | Additional recession coefficient of upper groundwater store | 0.05–0.99 |
| $K_1$ [d$^{-1}$] | Recession coefficient of upper groundwater store | 0.01–0.8 |
| $K_2$ [d$^{-1}$] | Recession coefficient of lower groundwater store | 0.001–0.15 |
| $L_P$ [-] | Soil moisture value above which actual evaporation reaches potential evaporation | 0.3–1 |
| $M_{MAXBAS}$ [d] | Length of equilateral triangular weighting function | 1–3 |
| $P_{MAX}$ [mm d$^{-1}$] | Maximum percolation to lower zone | 0–6 |
| $V_{UZL}$ [mm] | Threshold parameter for extra outflow from upper zone | 0–100 |

The runoff response function transforms excess water from the soil. This routine consists of upper and lower reservoirs that distribute the generated runoff over time. The lower reservoir is a simple linear reservoir representing a contribution to baseflow. It

also includes the effects of direct precipitation and evaporation over open water bodies in the basin. The lower reservoir storage, $S_{LZ}$ [mm], is filled by percolation from the upper reservoir ($P_{MAX}$), and the outflow from this lower reservoir ($Q_2$) is controlled by the recession coefficient $K_2$ [d$^{-1}$]. However, the upper reservoir storage $S_{UZ}$ [mm] is drained by two recession coefficients, $K_0$ [d$^{-1}$] and $K_1$ [d$^{-1}$], draining the quick flow $Q_0$ [mm d$^{-1}$] and slow flow component $Q_1$ [mm d$^{-1}$] separated by a threshold $V_{UZL}$ [mm] (Equations (3)–(5)).

$$Q_0 = K_0(S_{UZ} - V_{UZL}) \tag{3}$$

$$Q_1 = K_1(S_{UZ}) \tag{4}$$

$$Q_2 = K_2(S_{LZ}) \tag{5}$$

If the yield d$Q$ [mm d$^{-1}$] from the soil moisture routine exceeds the capacity, the upper reservoir will start to fill. This reservoir models the response during flood periods. The parameters calibrated from the runoff response function are $P_{MAX}$, $K_0$, $K_1$, $K_2$, and $V_{UZL}$. Finally, the runoff is computed independently for each sub-basin by adding the contributions from the upper and the lower reservoir. To account for the damping of the runoff pulse in the river before reaching the basin outlet, a simple routing transformation is performed. This filter has a triangular distribution of weights with the base length and is expressed by the parameter $M_{MAXBAS}$ [d]. A detailed description of the model is shown by Bergstrom [56] and Seibert and Vis [49]. The ranges of the nine model parameters are derived from prior knowledge, provided through a global set of HBV model parameters [30].

### 3.2. Parameter Estimation in the Gauged Catchments

The study used a uniform random sampling approach to produce simulated streamflow ensembles with the HBV model. To obtain reasonable parameter sets, 20,000 combinations of the nine HBV parameters were generated from a uniform random Monte-Carlo sampling procedure. There is a possibility to use other parameter optimization options. However, the aim of this study is not only to obtain the best parameter but also a set of large behavioral parameters that are later used to derive weights in the weighted regression procedure using the inverse of their coefficient of variation. The random sampling procedure has been effectively used in obtaining behavioral parameters and deriving parameter uncertainties for the previous regionalization studies [10,25]. Other global optimization procedures such as Dynamically Dimensioned Search and Shuffled Complex Evolution can find the best parameter sets [57]; however, the purpose of this study is to find a range of suitable behavioral parameters in addition to the best parameters. The choice of a uniform random sample is relatively computationally efficient and easy to implement. Therefore, the Monte Carlo method was selected to produce a large number of suitable behavioral parameters from the reduced parameter samples using a minimum 0.5 NSE threshold.

A split sample test [58] is used by splitting the simulation period into the calibration period (1995–2002) and the validation period (2003–2007) and calculating the Nash Sutcliffe Efficiency (NSE) [59] for both periods (Equation (6)):

$$NSE = 1 - \frac{\sum_{i=1}^{n}(Q_{obs} - Q_{sim})^2}{\sum_{i=1}^{n}(Q_{obs} - \overline{Q}_{0bs})^2} \tag{6}$$

where $Q_{obs}$ and $Q_{sim}$ are monthly averages of observed and simulated discharges [m$^3$ month$^{-1}$], respectively, while $\overline{Q}_{0bs}$ is the mean observed discharge over the calibration or validation periods. Using monthly averages, it is focused on the seasonal, long-term behavior instead of daily, short-term fluctuations. In order to remove unrealistic parameter combinations, only parameter sets that produced NSE $\geq$ 0.5 in the calibration period were kept. Consequently, different catchments can result in a different number of behavioral parameter sets. A pre-analysis using NSE on a monthly time scale showed that the model

performs well for all of the catchments. Comparing the mean and variability of model performance for the remaining catchments during calibration and validation allows to assess the predictive performance and uncertainty of the selected parameter sets. To prepare for regionalization, (1) the variability of each model parameter in the reduced parameter sample after NSE $\geq$ 0.5 (expressed by their coefficient of variation, CV), and (2) the best parameter set (largest NSE) of the calibration, $\text{NSE}_{\text{CAL}}$, and the validation period, $\text{NSE}_{\text{VAL}}$, for each of the catchments were extracted. We derived the behavioral parameter ranges (parRANGE) during calibration with NSE $\geq$ 0.5. Using the behavioral parameter sets, we ran the model for the validation period (2003–2007), and we selected the best validation parameter sets among the behavioral parameters. Among the behavioral parameter sets, there may be a best parameter set that performs differently from the best-calibrated parameter sets in some catchments. In addition, the most stable parameter set for each catchment was identified, i.e., the parameter set that showed the smallest difference in NSE values between calibration and validation periods, $\text{NSE}_{\text{DIFF}}$. Again, it is considered that the parameter set with the highest NSE is selected. In this procedure, after sorting the $\text{NSE}_{\text{DIFF}}$ in ascending order, only 5% of the parameters with the lowest $\text{NSE}_{\text{DIFF}}$ were extracted, from which the parameter set with the highest NSE was selected. In the selection of the stable parameter sets, parameter sets of the calibration period with the highest NSE that show a minimum difference with the NSE of the validation period were preferred.

### 3.3. Parameter Estimation in the Ungauged Catchments

To estimate model parameters for the 35 ungauged catchments, a parameter regionalization procedure was developed using weighted linear regressions (Equations (7)–(9)), also known as weighted least squares. This will allow linking the catchment properties (Table 2) and the estimated model parameters of the gauged catchments (see Section 3.2). Compared to the ordinary least squares, the weighted linear regression introduces a weight matrix to account for the unequal variances of observations [10,31]. This brings advantages in regionalization since the identifiability of a model parameter can vary significantly between catchments. To obtain a robust regression model, more weights can be put to the catchments with identifiable parameters. The weighted linear regression is described as follows:

$$Y = X\beta + \varepsilon \tag{7}$$

$$W = \begin{bmatrix} w_1 & 0 & \cdots & 0 \\ 0 & w_2 & \cdots & 0 \\ \vdots & \vdots & \ddots & 0 \\ 0 & 0 & \cdots & w_n \end{bmatrix} \tag{8}$$

$$\hat{\beta} = \underset{\beta}{argmin} \sum_{i=1}^{n} \varepsilon_i^2 = \left( X^T W X \right)^{-1} X^T W Y \tag{9}$$

where $Y$ and $X$ are, respectively, the response variable (estimated model parameters in this study) and the independent variable (catchment properties in this study). $\varepsilon$ is the error vector and W is a diagonal matrix containing weights. $\beta$ represents the regression coefficients vector and is estimated by $\hat{\beta}$, which minimizes the weighted sum of errors. The coefficient of variation (CV), which is the standard deviation divided by the mean of behavioral parameter sets of a catchment, represents the variability of a parameter. The smaller the CV, the less variable and more identifiable the parameter. Therefore, the reciprocal CV of the parameter of interest was used as weights for each catchment. Normally the weights in a weighted least squares regression are 1/variance. However, using 1/CV to calculate the weights brings the advantages of comparing the weights of a catchment for different parameters, because using 1/CV removes the influence of magnitude and units of a parameter. Additionally, using a weight 1/CV would result in a

similar result as 1/variance since a parameter introducing a constant (the mean) into the regression will not change the relative weights to each catchment.

A correlation analysis was performed between the model parameters and catchment properties to select the independent variables. With this regard, the linear correlation, Spearman's rank correlation [60], and the correlation on the log-transformed scale were applied. For the weighted linear regression, the catchment properties with the strongest correlation with the model parameter were chosen as the independent variable as shown in Table S1 (Supplementary Material). The normal scale linear regression model was selected over the log-transformed ones due to the better performance of the normal correlation than the log-transformed correlation. There is a possibility to do linear regression on the log-transformed scale; however, the correlation coefficients are superior on the normal scale to the log-transformed ones which can provide a better regression model on the normal scale. Choosing a log transformation reduces the weight of extreme values in predictor and response variables for large sample studies [15]; however, this may not be the case for the low catchment numbers. To increase the representation of more identifiable catchments, a weighted regression was applied on the normal scale. Parameters poorly correlated with the catchment property and remain unidentifiable throughout the catchments are rejected from the regression model, and their median value was considered as the best model instead.

### 3.4. Evaluation and Uncertainty Estimation of the Regionalization Procedure

A leave-one-out cross-validation method [61] was used for parameter estimation and for evaluating the prediction skill of the regressions. Leave-one-out is a simple cross-validation procedure: each regression model is created by taking all the catchments except one, the evaluation catchment. For large sample training data, this method is more resilient to irreducible errors [62]. Recently, this technique is applied for regionalization studies involving the prediction of discharge signatures [15,18,63] and model parameters in the ungauged catchments [7,8,64]. Thus, for 14 catchments, there are 14 different regression models and 14 different evaluation catchments. For each of the 14 iterations, the method produces one regionalized model parameter set for the left-out catchment using the regression model derived from the remaining 13 catchments and parameter sets. Since it is not known which parameter sets provide the most stable regionalization method, the procedure was repeated three times using the best parameter sets of (1) the calibration period, (2) the validation period, and (3) the most stable parameter sets between the calibration and validation periods. Thus, the quality of the regionalization procedure was evaluated three times. In order to choose the best of them for the following analyses, the simulations of the left-out catchments were evaluated for both the calibration (1995–2002) and the validation (2003–2007) period. In order to quantify the uncertainty of the regionalization procedure, all the 14 regionalized models were applied that were created for the leave-one-out evaluation to the ungauged catchments. Hence, an ensemble of 14 predicted streamflow time series is produced for each ungauged catchment to reflect the regionalization uncertainty.

### 3.5. Estimation of Regional Resilience of Streamflow to Precipitation Variability

In addition to the simulations at the 14 gauged catchments (and their respective uncertainty), the regionalization procedure produces simulations, and uncertainty estimates, of the 35 ungauged catchments, altogether covering a majority of the RVLB area (60.81%). The regional simulation tool was used to estimate the region's streamflow resilience to precipitation variability, which is quantified through streamflow elasticity [65]. The elasticity of streamflow quantifies the sensitivity of a catchment's streamflow response to the precipitation changes at the annual scale. Resilient catchments show low streamflow variability in response to precipitation changes. Elasticity is defined as the ratio between the

change in the annual aggregated discharge (*dQ*) and the change in the annual aggregated precipitation (*dP*) shown by Equation (10):

$$\varepsilon P = median\left(\frac{dQP}{dPQ}\right) \qquad (10)$$

The elasticity values were calculated for each year using the entire ensemble of simulated daily discharge time series at all gauged and ungauged catchments. From this, the median, wettest, and driest year elasticities were extracted for each catchment, including their uncertainty expressed by the respective CV resulting from all parameter sets with NSE $\geq$ 0.5 for the gauged catchments and the 14 parameter sets obtained through the leave-one-out cross-validation procedure for the ungauged catchments. The wettest year elasticity is calculated for the transition from the normal year to the wettest year, and the driest year elasticity is calculated for the transition from the normal year to the driest year. Using the median, wettest, and driest year elasticities, it is possible to learn how resilient a hydrological system is against extreme climatic conditions such that one can evaluate the corresponding security of the water supply. For instance, if a catchment has a large streamflow elasticity, there will be a large system change in response to a big change in precipitation (e.g., the driest year); such a system is not resilient as there would be a huge change in the water available, thus affecting the security of the water supply.

## 4. Results

### *4.1. Estimated Parameters in the Gauged Catchments*

The model calibration was performed using a split-sample test for the period 1995–2002, with a threshold of 0.5 NSE used to determine the behavioral parameter sets. Consequently, the model was run using the behavioral parameter sets for the validation period (2003–2007). Using the 0.5 NSE threshold, a wide range of behavioral parameter sets were obtained in all gauged catchments during calibration. The threshold-based separation of behavioral and non-behavioral parameter sets was used as a threshold-based approach, which helps to explicitly state under which minimum performance requirements, i.e., NSE $\geq$ 0.5, regionalization by the CV-weighted regression was conducted. Figure 3 shows the monthly NSE values for the calibration and validation periods, as well as their uncertainty, using behavioral parameter sets. For the monthly calibration, the model performance results in NSE 0.53 to 0.86 in the 14 gauged catchments. The maximum NSE values were highly reduced from the calibration to the validation periods for catchments #07 and #09. Catchments #01 and #03 show a small decrease in the monthly NSE values from the calibration to the validation period (Figure 3). In most cases, the monthly model calibration and validation show a uniform distribution of NSE values, indicating the stability of model parameters. The NSE values for the stable parameter set show the maximum value (0.80) for catchments #08 and #13, demonstrating good predictive skills in these catchments. Catchments #04 and #06 show lower parameter stability. The monthly NSE of the validation period was the highest for most catchments compared to the calibration period. This comes from the possibility to choose the best parameter from the confined parameter ranges using a threshold of 0.5 NSE throughout the catchments. The standard deviation of the monthly NSE values shows the ranges of uncertainty provided by the simulation for the selected parameter ranges (Table S3). In this regard, catchments #04, #05, and #06 result in the lowest simulation uncertainty during calibration.

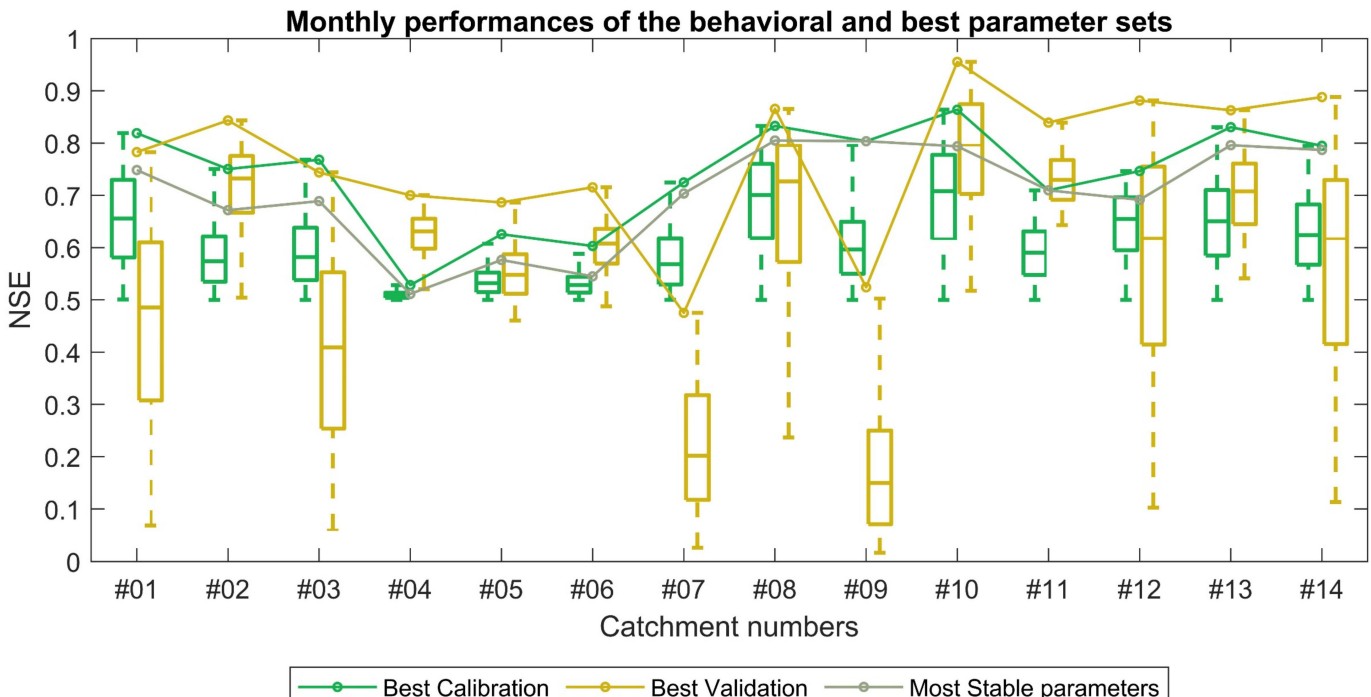

**Figure 3.** The ranges of monthly NSE values derived from the confined parameter sets during calibration and the corresponding NSE ranges during the validation period derived from the confined parameter sets; and the monthly NSE derived from parameter sets of best calibration, best validation, and most stable parameter sets for each catchment.

In addition to the uncertainties, Figure 4 shows the identifiability of the model parameters in the 14 catchments using the Cumulative Distribution Function (CDF). More deviation from the 1-1 line shows higher identifiability by the parameters. The catchments with highly identified parameters are shown by the dark gray color on the CDF. Figure S1 shows the variability of the confined model parameters obtained by the calibration, which is later applied to derive the weights of the regional regression model. The parameters that are well-identified are highly represented in the regional model by their weight (1/CV). In this regard, parameters such as $\beta$, $F_C$, $K_2$, and $L_P$ were well identified in most of the catchments. Parameter $\beta$ was highly identified towards the lower values as shown by the dark gray color (Figure 4) or by the narrow parameter range in catchment #6 (Figure S1). $F_C$ shows identifiability towards higher values in their parameter range for catchment #2. On the other hand, $K_2$ showed higher sensitivity towards the lower values for catchment #7. The confined ranges of model parameters ($K_0$, and $V_{UZL}$) result in a relatively uniform distribution in their median values for all catchments. These parameters ($K_0$ and $V_{UZL}$) remained insensitive and were later excluded from the regression procedure due to less information contained in their parameter identifiability. Taking median values for unidentifiable parameters would be a better model for the regional model [10] (Figure 4, Figures S2 and S3).

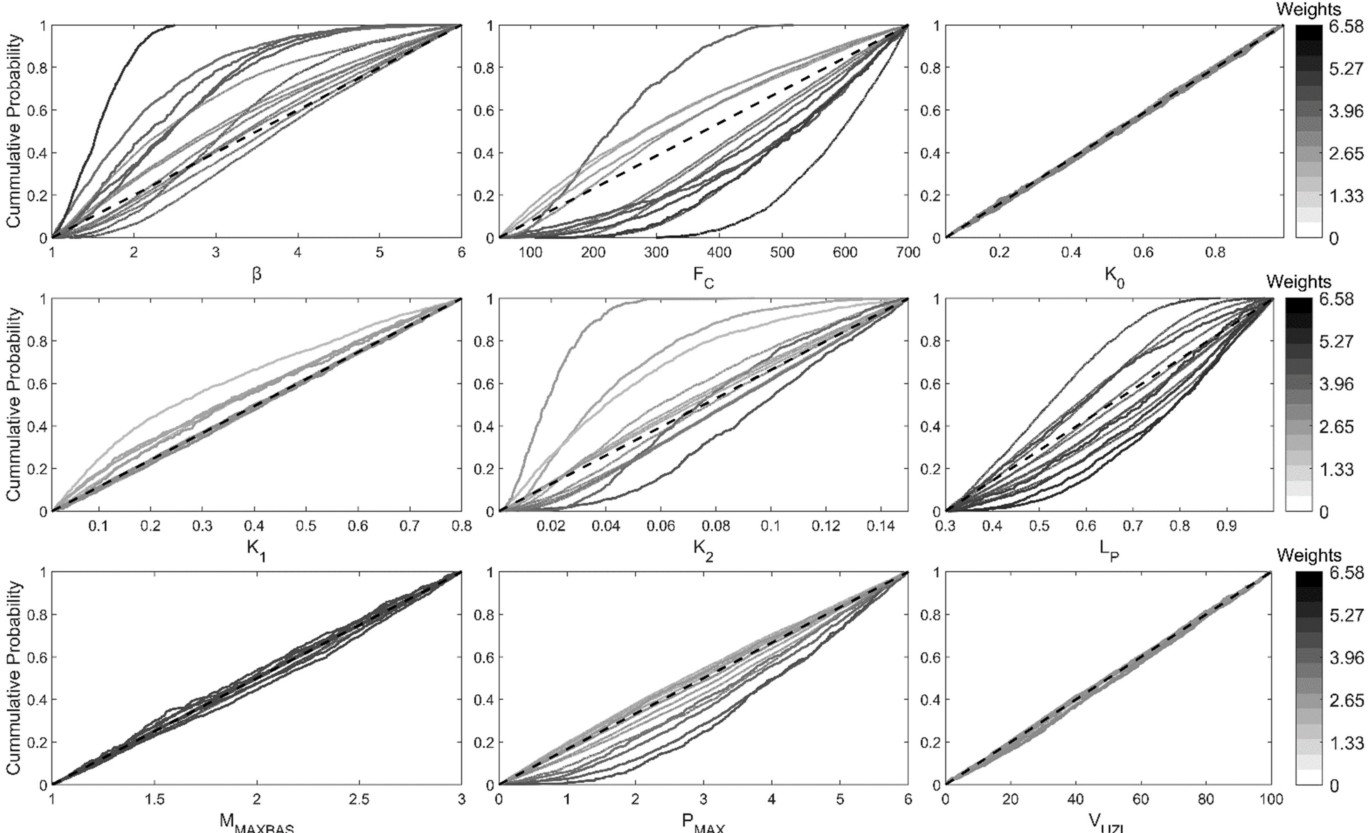

**Figure 4.** Cumulative distributions of parameters in the 14 catchments that show the probability of identifiability for each parameter on their ranges. The ranges between the most identifiable and poorly identified parameters are indicated through dark gray to light gray colors. The respective identifiability is also shown by the weight (1/CV) of parameter distribution. Where the dark gray color shows the catchment number with the most identifiable parameter. Parameters such as $K_0$ and $V_{UZL}$ are not identified through their parameter ranges, therefore the median of the parameter was used for the regression model.

### 4.2. Performance of the Regionalization Procedure

The transferability of the model parameters derived from weighted regression was evaluated using a leave-one-out cross-validation technique for each of the 14 gauged catchments. Figure 5 shows the results of evaluations for the three regionalized models on both calibration (1995–2002) and validation periods (2003–2007). Figure 5a,b shows the relationships between the performances of the monthly NSE of the best parameters estimated from calibration, validation, and the most stable sets and the corresponding regionalized parameter while evaluated on the calibration and validation periods. The regional parameter sets derived from the most stable parameters show the best performance in predicting discharge in the left-out catchments (NSE REGstable in Figure 5c) while evaluated during the validation period. The results show that 13 out of 14 catchments exceed a 0.25 NSE value and 10 out of 14 catchments exceed a 0.5 NSE value for the regression model derived from the most stable parameters during evaluation in the calibration period (1995–2002) (Figure 5a). Furthermore, the median NSE value of the 14 catchments for the regression model derived from the most stable parameters (NSE REGstable) is 0.63, compared to 0.50 for the best-calibration ones (NSE REGcal) and 0.57 for the best-validated parameter sets (NSE REGval) (Figure 5c) for the evaluation during the validation period (2003–2007). To assess the influence of low flow and high flow scales on the regionalization performance, the NSE was calculated on the Log transformed scale (logNSE) of observed and simulated discharge. Table S4 shows the logNSE values during the evaluation of the regression model

on both the calibration and validation periods. The median NSE values were not improved on the log scale compared to the normal scale evaluation (Table S4).

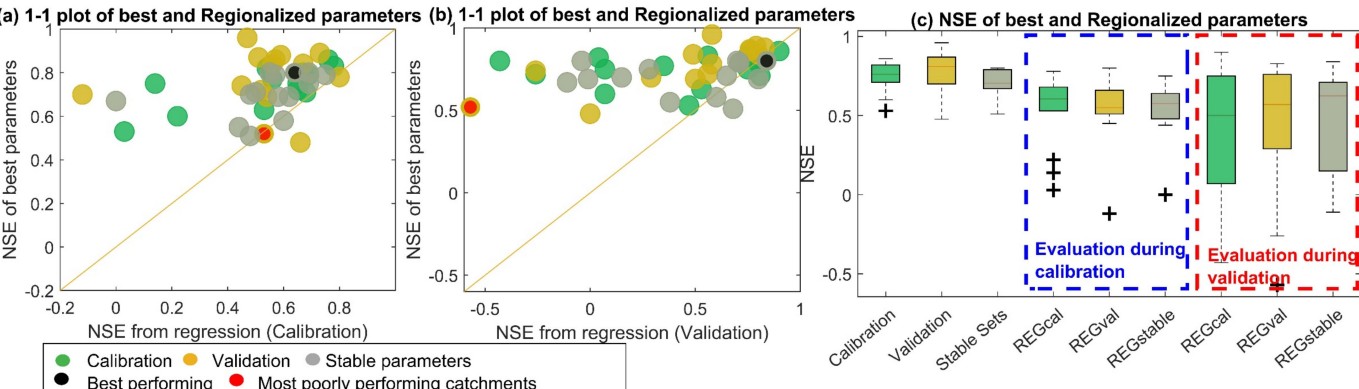

**Figure 5.** Evaluation of the regression procedure on the calibration (1995–2002) and validation period (2003–2007), showing (**a**,**b**) the relationship of the NSE of the best-estimated parameters (from calibration, validation, and stable parameters set) and the parameters from the regression. The black and red scatter indicate the best and most poorly performing catchments, respectively. (**c**) Shows the performances of best-estimated parameters (from calibration, validation, and stable sets) and the performance of the three regression models developed using the best parameters of calibration (REGcal), validation (REGval), and stable sets (REGstable) while evaluated during the calibration and validation period.

Figure 6 shows the relationships between the best model parameter set obtained from the most stable parameter and the regression for the 14 catchments. However, these results should be carefully interpreted for data-scarce regions due to the use of available global data sets (precipitation and potential evapotranspiration). Parameters $K_0$ and $V_{UZL}$ were unidentifiable during calibration, and their median values were taken for the regionalized model. The weighted regression shows acceptable performance in reproducing most of the parameters. For instance, the weighted regression reproduces well the parameters $\beta$, $F_C$, and $K_2$, whereas the remaining parameters are less reproducible by the regression model. Table S5 shows the performance of the weighted regression model during the leave-one-out cross-validation using a coefficient of determination ($R^2$). It is seen that the weighted regression procedure does not always produce model parameters in their predefined range (Table 3). For example, the regressed $F_C$ of catchment #6 is above the maximum threshold of 700. In such cases, the outlier model parameter is assigned to its maximum value. Among the remaining parameters, most of them show acceptable correlations. However, some of them, such as $K_1$ and $P_{MAX}$, are poorly reflected through the regional regression in a few catchments. For instance, $K_1$ in catchment #11 is poorly modeled. Therefore, parameter $K_1$ is not identified in catchment #11. The regression model using parameter $P_{MAX}$ is poorly represented in four catchments (#09, #10, #13, and #14). This shows that for catchments #09, #10, #13, and #14, the parameter $P_{MAX}$ is not identifiable for the carefully selected catchment properties. Furthermore, for catchment #09, model parameters $F_C$, and $P_{MAX}$ were poorly identified. However, the weighted regression procedure sufficiently represents the parameters in the remaining catchments. The best performing (#08) and most poorly performing (#09) catchments were also shown by the black and red colors, respectively. Catchment #08 primarily shows a stable prediction for all parameter-sampling procedures during calibration, validation, and stable relationships. Identifiable parameters in this catchment are also reproduced well from the regression model, whereas in catchment #09 (red scatter), most parameters were poorly identified and poorly reproduced by the regression model.

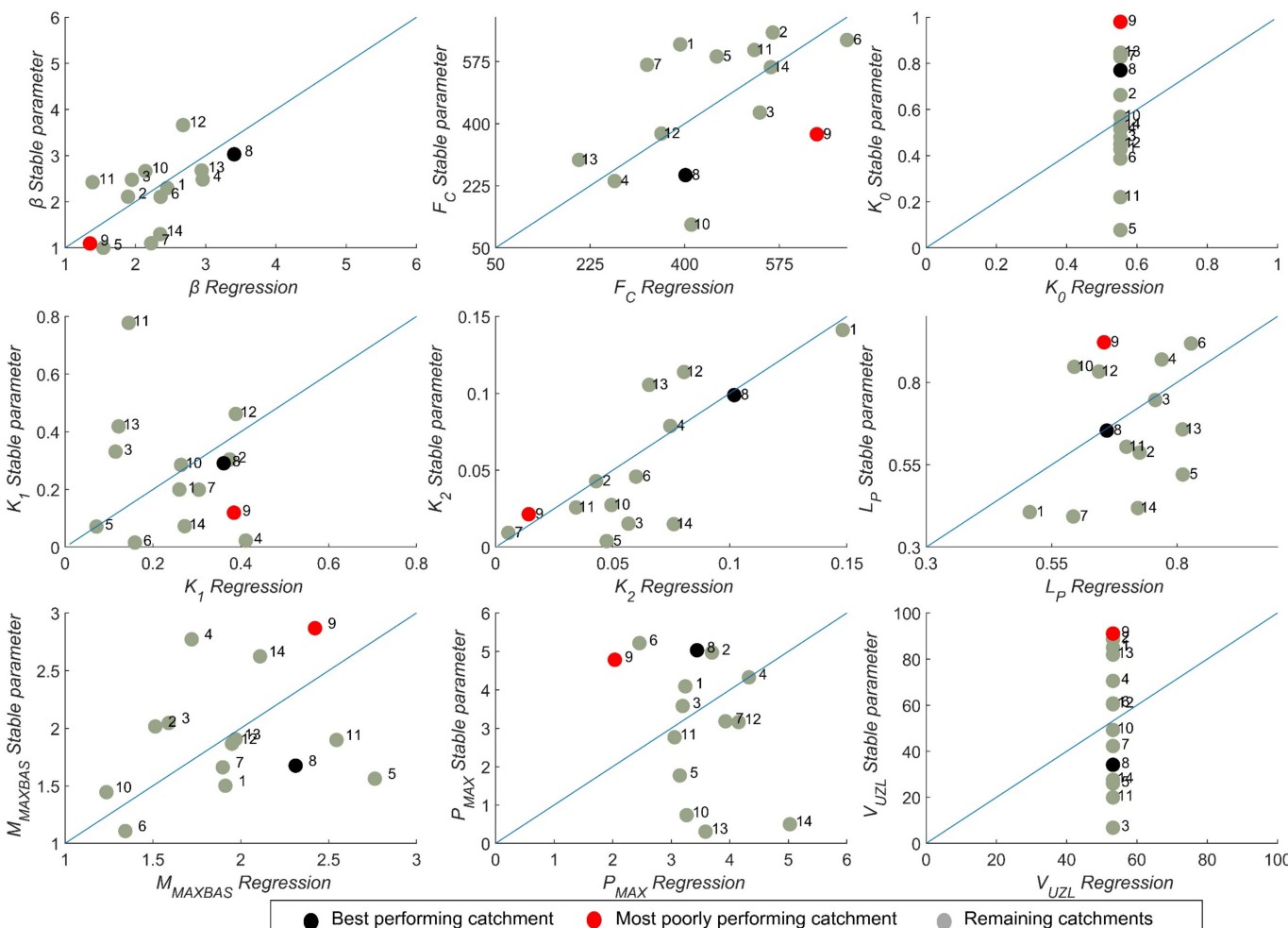

**Figure 6.** Relationships between best parameters estimated from stable sets and parameters derived from the regression model. The black scatters (catchment #8) represent the catchment with most parameters that are highly identified during parameter estimation and its corresponding parameters during regression. The red scatters (catchment #9) represent a catchment with most parameters that are poorly identified during parameter estimation and its corresponding parameters during regression. For unidentifiable parameters ($K_0$ and $V_{UZL}$), their median value was taken for the regionalized model.

The reliability of this approach is further evaluated by comparing the observed discharge with the uncertainty interval of the regionalized model while the model is running for the validation period (Figure 7). The observed discharge for the best-performing catchment (#08) is enveloped by the prediction interval during the low-flow and high-flow periods. Furthermore, it corresponds highly with the mean of the 14 regionalized simulations (Figure 7a), whereas the observed discharge for the most poorly performing catchment (#09) is not well captured by the prediction interval nor by the mean of the 14 regionalized simulations (Figure 7b).

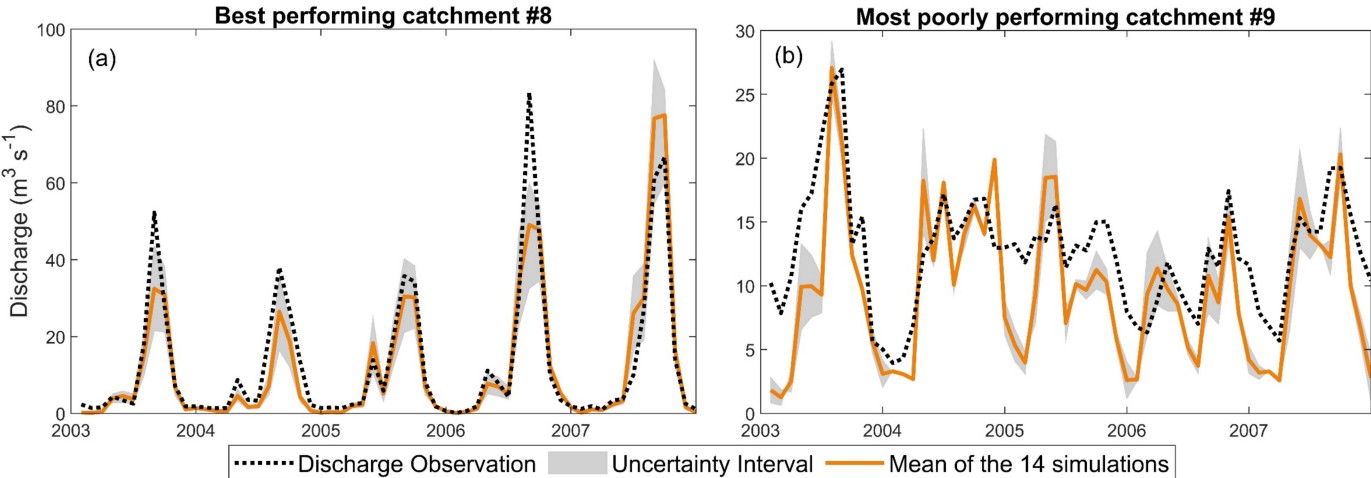

**Figure 7.** Prediction interval derived from the 14 regression models using the best parameter sets from stable parameter sets and uncertainty interval; (**a**) for best−performing catchment (#08) and mean of the 14 simulations, and (**b**) uncertainty interval for most poorly performing catchment (#09) and mean of the 14 simulations.

### 4.3. Estimation of Regional Resilience of Streamflow to Precipitation Variability

With the acceptable performance of the regional regression model, it has been applied for the ungauged catchments over the entire RVLB region. Figure 8 shows the median, wettest, and driest year elasticities computed from both gauged (all parameter combinations with NSE $\geq 0.5$) and ungauged catchments (all 14 regionalized parameter sets from the leave-one-out cross-validation). The median elasticity values are fairly distributed in the basin. However, the highest elasticity is shown in the western part of RVLB (Figure 8a). For the wettest year, the highest elasticity results from the catchments in the south and the northeast show less resilience to extreme precipitation (Figure 8b). Relatively low elasticities result from the driest year, except for a few catchments in the south. This variation is because catchments in the south, except for some outliers, are mainly dry and receive a comparatively low amount of precipitation.

The uncertainty of the 14 ensembles derived from the regression was shown using the CV. It is seen that the median values, as well as the wettest and driest year elasticities, show low uncertainty in the gauged catchments, indicated by low CVs (Figure 8d–f) compared to the ungauged regions, whereas the highest CV values are shown for the median, wettest, and driest years for the ungauged catchments in the southern region. However, most of the catchments in the central and northern parts show less variability, which shows low uncertainties for these catchments. The highest and lowest precipitation will eventually result in different values of elasticities. The differences in the catchment properties would result in the variability of the CV of the wettest and driest year for the neighboring catchments (Figure 8e,f). In areas such as the RVLB, the highest yearly and seasonal precipitation amounts could be interrupted by the seasonal or yearly dry spells [37]. The response to runoff is low in the driest year, as shown by the low elasticity values in most of the ungauged catchments (Figure 8c). Comparatively, the driest year CV in the southern part shows relatively lower values than the wettest year.

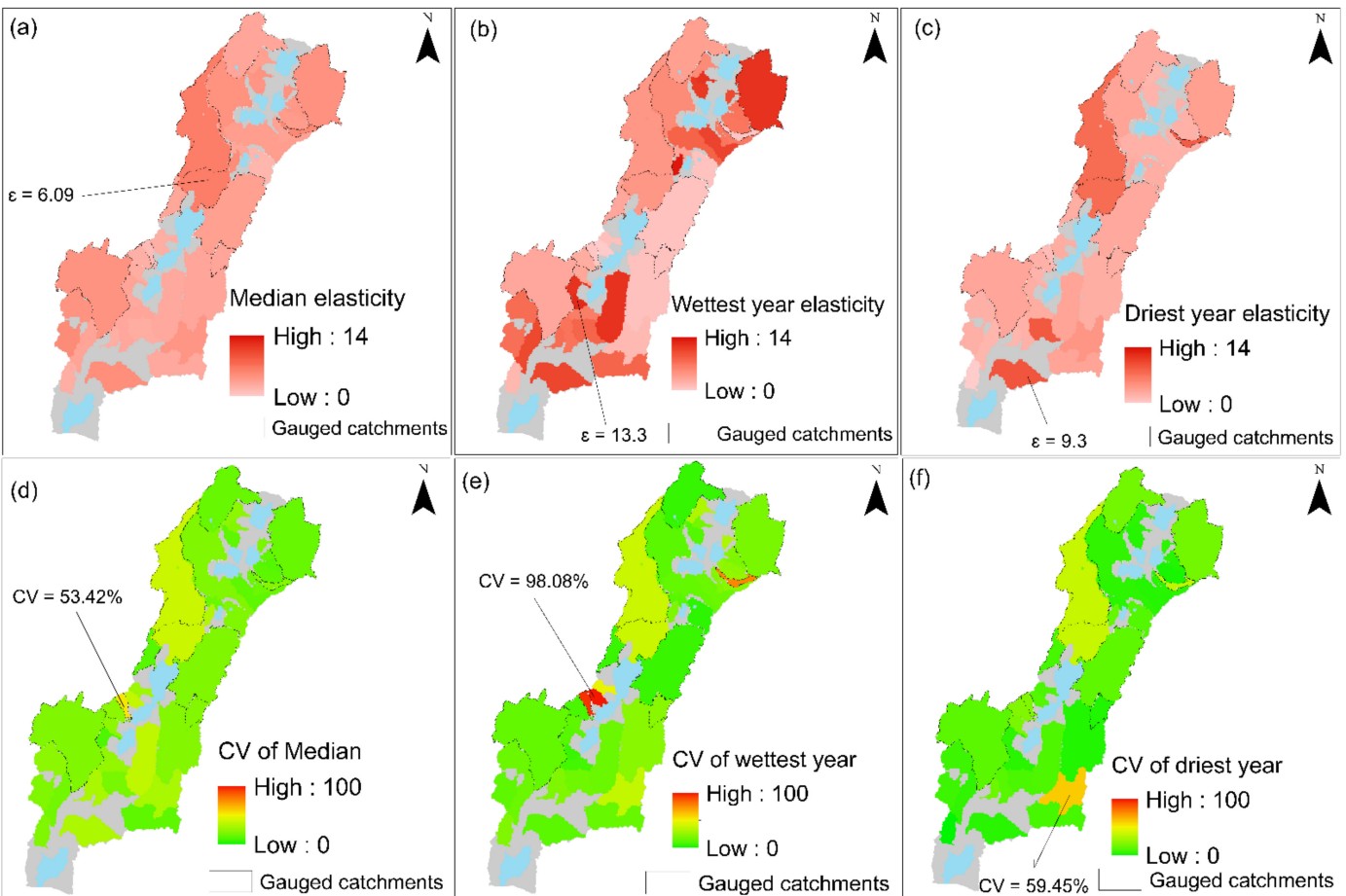

**Figure 8.** (**a**–**c**) Elasticity calculated for the median, wettest, and driest years for both gauged and ungauged catchments and (**d**–**f**) their corresponding simulation uncertainty expressed by the coefficient of variation CV.

## 5. Discussion

### 5.1. Reliability of the Regionalization Approach

The study demonstrates the applicability of a process-based hydrological model at a regional scale despite data scarcity. The reliability of the entire approach is shown through a three-step parameter estimation and model evaluation procedure, which enables one to identify the most reliable setting in the regional model development procedure. Overall, the stability of the parameters for the calibration and validation periods remained acceptable. Uncertainty was quantified by parameters sampled from the Monte Carlo random sampling procedure. The split sample test was applied to select the best parameter from the calibration and validation periods. Furthermore, the most stable parameter sets were calculated from the calibration and validation periods, which have also been performed in previous studies aiming at stable predictions [66]. The temporally stable model parameters indicate a relatively uniform system response during both calibration and validation and provide strong justification for the regional extrapolation of model parameters.

Parameters were estimated for the ungauged regions through the weighted regression procedure. Like previous studies using a similar approach [10], the weighting procedure increases the representation of identifiable parameters from the catchment in the regression model. More weight (reciprocal of CV) is assigned to the more identifiable parameters in a given catchment. Consequently, three regional models were derived using the best parameter sets obtained from the calibration, validation, and most stable parameter sets between calibration and validation for the 14 catchments. Figure 6 presents the nine parameters reproduced by using the most stable parameters, while the remaining two

regression models (using best calibration and validation) are shown in Figure S2 and Figure S3, respectively. Comparing the performance (NSE) of these three regression models (Figure 5c), a regional model derived from using the most stable parameter has shown superiority over the other two. These procedures increase the reliability of the approach for the model development. Uncertainty was also quantified through the leave-one-out spatial cross-validation of parameters. In this approach, every catchment is considered as an evaluation catchment, while a prediction is formed using the remaining catchments. Therefore, the 14 catchments produced a 14-regression model in the leave-one-out cross-validation method that quantifies prediction uncertainty in the ungauged catchments. The leave-one-out spatial cross-validations also show better performance in the regionalization studies that use discharge signatures [15,18] and for ungauged catchments parameter prediction [7,8,65]. This method is more stable and more resilient to irreducible errors for large sample studies [63]. The leave-one-out cross-validation approach also has a uniformly low bias and root mean squared error (RMSE) [62]. Therefore, the approach combines three steps of uncertainty quantification from (1) the parameter sampling, (2) the best parameter set identification, and (3) the leave-one-out cross-validation in the data-scarce regions.

A scatter plot of monthly NSE values between the parameters estimated (from the model calibration, validation, and stable sets) and the parameters regionalized from the regression equations shows the evaluation and reliability check. Figure 5b indicates that the median NSE of regression from the most stable parameters outperforms the NSE of parameters obtained from calibration and validation for leave-one-out cross-validation. Furthermore, it shows most of the scatter points are above the 0.5 NSE limit, indicating a better reproduction of parameters while evaluated during the calibration period (1995–2002). In this regard, the parameters in 71% of the catchments result in an NSE of $\geq 0.5$, and in 93% of the catchments, an NSE of $> 0.25$, for the regression model derived from the most stable parameters. The median value of the NSE for the 14 catchments is 0.63 during evaluation during the validation period (2003–2007); this is a sufficient performance in regionalization despite the few catchment numbers applied (Figure 5c). Studies showed difficulties in the regionalization performances for both the model parameters and discharge signatures [6,15,30,67]. With few training datasets used, the regionalization performances are acceptable which are derived from the monthly calibrated and validated parameters. The study focused on the challenges in regionalization in data-limited conditions showing the applicability of global forcing data for the regionalization of data-scarce regions considering the resulting uncertainties. This approach provides a basis for regional model estimation and uncertainty quantification for low catchment numbers in the data-scarce regions. The regionalization of the model parameters with the use of global data products in the data-scarce region can be expected to result in considerable uncertainty. However, this approach demonstrated that an acceptable median NSE value can be obtained despite low catchment numbers. In order to check the influence of low and high flow scales on the interpretation of the NSE, a more balanced logNSE to the regionalization performance was applied. Zhang and Post [68] show the influence of log transformation on the low-flow and high-flow scales. Table S4 shows the evaluation of the regression model on both the calibration and validation periods for the log-transformed observed and regionalized discharge. In this regard, the median NSE values have not improved the results any more than the normal scale NSE evaluation.

Figure 6 shows that more identifiable parameters for the gauged catchments (Figure 4) are also reproduced well from the regression model. The parameters of catchment #08 (black scatter) are highly identifiable during parameter estimation and the resulting parameters from the regression model are highly reproduced, whereas parameters in catchment #09 (red scatter) were poorly estimated and poorly reproduced by the regression model. Studies also show the selection of a more identifiable catchment as a donor for parameter regionalization depends on the score during the parameter calibration and validation [30].

In general, the evaluation indicates that despite ignoring parameter interactions [23,45,69], the regionalization procedure produces useful predictions of the model parameters in

the ungauged catchments. Plotting the simulated (monthly) time series of the best and the most poorly performing regional catchments (Figure 7), the study showed that the uncertainty estimates derived from the leave-one-out cross-validation procedure capture well the simulation uncertainty for the best-performing catchment #08. On the other hand, the prediction interval of catchment #09 shows less agreement with observed discharge, particularly for low flow and high flow, but with the estimated ranges of uncertainty much closer to the observations. Previous regionalization studies have already shown that some outlier catchments cannot be captured if they behave very differently from the general trend [25], as is most probably the case for catchment #09. However, since most regionalized catchments are with an NSE of $\geq 0.5$, there is sufficient reason to believe that the parameters of most ungauged catchments in the RVLB can be approximated acceptably.

*5.2. Parameter Sensitivity and Spatial Variability*

It is found that the model parameters have different sensitivities in the gauged catchments (Figure 4). Since the hydrogeology of the region is very heterogeneous, the parameters controlling the underground water flow show high variabilities among the catchments. Volume-controlling parameters ($\beta$, $F_C$, and $L_P$) and the recession coefficient in the lower reservoir ($K_2$) are readily identifiable in most catchments. However, the insensitivity of the parameters in some catchments may be due to interactions with other parameters. Abebe et al. [70] show the interaction between parameter $K_2$ and the percolation rate ($P_{MAX}$), where the increment of $K_2$ beyond the optimum rate of percolation may not show any sensitivity. In addition, the insensitivity of model parameters is related to the poor identifiability of model parameters in the catchments. For instance, parameters such as $K_0$ and $V_{UZL}$ were poorly identified and remained insensitive for any parameter value throughout the catchment (Figure 4).

The most identifiable parameters in a catchment result in the highest sensitivity towards any parameter value. However, parameter insensitivity is due to variations in the values of catchment properties. This is illustrated by parameter $K_2$, which is a recession coefficient in the lower reservoir, and it is well identified in catchments #07 towards the lower parameter value. Catchment #07 is characterized by sloppy catchment with a high drainage channel slope (catchment index) (Table 2) which facilitates the drainage of available water by surface runoff. Thus, for the little remaining water in the lower reservoir, lower values of $K_2$ will become more sensitive. Parameter $K_1$, which is a recession coefficient from the upper reservoir, is well identified in catchment #05. A lower value of the recession coefficient results in a relatively low drainage density in catchment #05 (Table 2), which facilitates a relatively higher percolation ($P_{MAX}$) in the catchment.

The interactions among model parameters may not be the only reason for insensitivity as it varies in different catchments and their properties. For instance, sloppy catchments in a small drainage area (#07 and #09) facilitate the conversion of precipitation into runoff, resulting in less soil moisture in the upper reservoirs in the HBV model. In such cases, adjustment to a parameter ($K_1$) controlling the water flow might not affect the outflow conditions. This is also shown by the negative correlation of the slope with $K_1$ (Table S1). Furthermore, the insensitivity of the parameters in the upper reservoir can be affected by low precipitation amounts. In low precipitation conditions (such as in catchments #08, #10, and #13) the resulting soil moisture from the soil profile and the upper reservoir will be much less, resulting in less runoff. The adjustment of the runoff-controlling parameters ($K_0$ and $V_{UZL}$) might not have any influence (remain insensitive) on the resulting runoff. Parameter $K_0$ only functions when the cumulative precipitation exceeds the threshold of the $V_{UZL}$ value. Other than climatic properties, the insensitivity of the parameters can result from the interaction of the parameters, and this influence will be more pronounced for complex models. Other studies also showed that the insensitivities of the model parameters could result from the mismatch between the model complexity and the available data used to parameterize the model [71–74].

*5.3. Estimation of Regional Resilience of Streamflow to Precipitation Variability*

In this study, elasticity values for the median, wettest, and driest years and their respective CVs were calculated (Figure 8). In the regions with higher elasticities (Figure 8a–c), the catchments respond faster to any change in precipitation promoting fast flow components. However, for low elasticity values, the streamflow responds slowly to precipitation change.

The CV indicates the uncertainty of prediction for the gauged and ungauged catchments (Figure 8d–f). Most of the catchments located in the southern part of the basin show a higher CV value in combination with a low resilience of streamflow to precipitation variability. Prediction variability is higher in the southern part compared to the north, and there are also few gauged catchments. Therefore, the higher uncertainty in the south may be attributable to the mixed effects of higher precipitation variability and the remoteness of the gauged catchments used to establish the regional regression.

The variability in areas of gauged and ungauged catchments used for parameter estimation and prediction, respectively, reduces with the strength of correlation between calibrated parameters and catchment properties (Table S1). Other studies also show that runoff from smaller catchments can have a stronger relationship with local climate and catchment properties than larger catchments [24,45,75]. In addition, the wettest year shows higher uncertainty compared to the driest year. However, most of the ungauged catchments that are located in the northern parts show lower uncertainty, more or less similar to the gauged catchments. This is because these ungauged catchments are much closer to the gauged catchments and are hence better represented by the regression model than the remote ungauged catchments. This is supported by the study of Qi et al. [76] that compared different regionalization techniques and found the strongest performance using the spatial proximity under different climate regions. Moreover, most of the streamflow in the north remains more resilient to precipitation change as it shows lower elasticity values, which indicate a higher resilience to streamflow in the dry years (Figure 8c). Furthermore, the CV for the driest year shows a relatively low uncertainty in the northern and eastern parts for both gauged and ungauged catchments.

The application of this approach to the RVLB shows that the predicted elasticities are characterized by a wide range of uncertainty for the ungauged catchments in the southern part. The reasons for such variation in uncertainty could be the fact that the ungauged catchments in the southern part are mainly dry and receive a comparatively low amount of rainfall (Table S2). This is different from the wetter, northern part, where most of the gauged catchments are located. This accords with studies showing a decreasing regionalization performance from smaller and more arid catchments [77]. Previous work has also shown that the spatial variability of precipitation can interact with catchment properties to alter hydrological processes [78]. Thus, higher precipitation variability in the gauged and ungauged catchments introduces more uncertainty to parameter regionalization. Such variability in prediction also results from the relative location of an ungauged catchment to the gauged one where the regional model is developed [79].

This approach shows variability in the resilience of gauged and ungauged regions, which emerges from parameter uncertainty and climate variability. Over the RVLB, lakes are particularly stressed by growing water demand, climate variability, and drought [39]. The reliability of open water resources in low-resilient catchments remains uncertain. Coupled with the significant reduction in lake sizes and water levels [80], this will negatively affect water resource availability and ecosystem stability in the future.

*5.4. Transferability of the Approach to Other Catchments and Models*

This study provides a methodology that accounts for uncertainty throughout all steps of the regionalization approach. It translates data limitations into the remaining uncertainties that were found in the regional simulations (expressed, for instance, by the CV values of the elasticities). The approach is independent of the model and the number of parameters, but it is expected that a more complex model (with more parameters)

would struggle more through data limitations and generate larger uncertainties when applied regionally.

In this study, the identifiability of the model parameters varied across the 14 catchments. This could be from the variation in the catchment properties used for the regionalization. Van Esse et al. [81] show the performance of conceptual hydrological models to vary depending on the size of a catchment and the soil moisture state of a catchment. Yang et al. [73] also showed the regionalized model to be dependent on the complexity of hydrological models in different climatic regions. In this approach, the variation is accounted for through regionalization uncertainties by applying the leave-one-out spatial cross-validation. Thereby, one can have different levels of parameter identifiability in the transferability of a model parameter from one model component to another. By choosing stable parameter sets for the regional model, one can guarantee the transferability to another region by the spatially evaluated robust regional model within their uncertainty range. Nevertheless, model parameter transferability does not always result in success. The difference in model parameters could derive from the complex relationship between different model components within the model structures. The transferability of insensitive parameters ($K_0$, and $V_{UZL}$) could add bias to the regionalized models, thus taking their median values from the estimated parameters can be a better model [10]. The stability and resilience of the model parameters from the regionalization procedure would minimize the error and bias in parameter transfer within the model component [62]. However, success in the parameter transfers will be influenced by the dominant catchment properties that are identified regionally [32]. Despite this, a sufficient correlation between the catchment property and model parameters can be a good indicator of the predictive power of the selected catchment properties.

## 6. Conclusions

The study demonstrated the use of global data products for the regionalization of model parameters using a small sample of gauged catchments despite data scarcity. The study applied multiple options for parameter estimation and model evaluation procedures. Three steps of uncertainty quantification were combined from the parameter sampling, best parameter sets identification, and leave-one-out cross-validation. The study demonstrated the validity and reliability of this approach at 14 test catchments with varying catchment properties. The parameter estimates from the leave-one-out cross-validation using the most stable parameters have outperformed the parameter estimates from best-calibrated and best-validated parameter sets.

The study incorporated uncertainties from the leave-one-out cross-validation that can provide a robust way of uncertainty quantification by generating 14 estimates of plausible streamflow ensembles and simulation uncertainties. This approach shows variability in the resilience of gauged and ungauged regions, which emerges from parameter uncertainty and climate variability. The study showed the uncertainties of elasticities in the gauged catchments obtained from simulation to be less than that of the uncertainties of elasticities in the ungauged catchments obtained from regionalization. The study further enables the quantification of the wettest and driest year elasticities and their uncertainties throughout the catchments, which provides a basis for the integrated water resource management in the region. Linking this approach with more observations of catchment properties on a larger scale can provide a good basis for large-scale water resource management. This approach can be extended to simulate and quantify the resilience of gauged and ungauged regions under climate change by quantifying the additional uncertainties emerging from climate projections.

Overall, this approach provided directions for uncertainty reduction by combining global input data with local discharge measurements, which result in a refinement of estimated model parameters for both gauged and ungauged catchments. The acceptable monthly simulation results obtained in the gauged catchments and the acceptable results of the spatial split-sample test (with median Nash Sutcliffe efficiencies of 0.63) indicate

that global products can be used as model inputs to provide reasonable simulations in data-scarce regions. Small-scale studies that use simple models (few parameter numbers) show the transferability of the model parameters to ungauged basins [10]. In extension to this, this approach provides the possibility of identifying parameters of ungauged basins in data-scarce regions, including a thorough evaluation and uncertainty quantification procedure. Using well-identified parameters, more reliable regional relationships can be obtained from the most dominant catchment attributes. In either large or low samples of catchments, a leave-one-out procedure should always be possible. By applying a specific hydrological model (HBV) and using a general framework, it can be also easily adapted to other hydrologic models. This approach demonstrated a way for robust parameter regionalization through leave-one-out cross-validation that enables the transfer of parameters across regions. It is proposed for further research that this approach can be applied for parameter transfer with model structures other than HBV to elaborate its general applicability in data-scarce regions. As this approach is model-independent and the input data used are available globally, it can be applied to any other data-scarce region where predictions of regional water availability are required. Despite the results, caution should always be taken when considering low catchment numbers as poorly assessed catchment properties or imprecise forcing data for some of them can greatly bias regional estimates.

**Supplementary Materials:** The following supporting information can be downloaded at: https://www.mdpi.com/article/10.3390/hydrology9080150/s1. Table S1 Correlation coefficients (CC) between catchment properties and model parameters. Table S2 Catchment properties that are derived for streamflow prediction in the ungauged catchments using the 14 parameter sets obtained by the weighted regression. Table S3 The maximum NSE on the monthly scale for the best parameters derived from calibration, validation, and stable sets, and standard deviations (std) of monthly NSE for calibration and validation. Table S4 Evaluation of the three different regression models from calibration, validation, and stable sets on both calibration (1995–2002) and validation periods (2003–2007). NSE was calculated on the Log transformed scale of observed and simulated discharge. Table S5 R2 of weighted regression performance for the nine HBV parameters during the leave-one-out cross-validation. Figure S1 Confined ranges of model parameters derived during model calibration for each catchment. Figure S2 Scatter plots between best parameters estimated from calibration and parameters derived from the regression model. The black circle (catchment #8) represents best performing catchment from parameter estimation and its corresponding parameters from regression. The red circle (catchment #9) represents the most poorly performing catchment by the regression model. For unidentifiable parameters (K0 and VUZL) their median value was taken for the regionalized model. Figure S3 Scatter plots between best parameters estimated from validation and parameters derived from the regression model. The black circle (catchment #8) represents best performing catchment from parameter estimation and its corresponding parameters from regression. The red circle (catchment #9) represents the most poorly performing catchment by the regression model. For unidentifiable parameters (K0 and VUZL) their median value was taken for the regionalized model.

**Author Contributions:** T.A. conceptualized and wrote the paper, developed and applied the regional model, and analyzed and visualized the results. Y.L., S.T. and A.H. provided supervision and advice throughout the development and application of this study and provided support in developing the manuscript. All authors have read and agreed to the published version of the manuscript.

**Funding:** The German Academic Exchange Service DAAD and Hawassa University under the EECBP Home Grown Ph.D. Scholarship Programme, 2019 (57472170) supported Tesfalem Abraham. Yan Liu and Andreas Hartmann were supported by the Emmy-Noether-Programme of the German Research Foundation (DFG, grant number: HA 8113/1-1; project "Global Assessment of Water Stress in Karst Regions in a Changing World").

**Institutional Review Board Statement:** Not applicable.

**Informed Consent Statement:** Not applicable.

**Data Availability Statement:** The climatic forcing data are publicly available and can be obtained via the link http://www.gloh2o.org/mswep for MSWEP (access date: 14 January 2020) and from https://www.gleam.eu for GLEAM (access date: 20 January 2020). The simulation of the hydrological model was conducted using a freely available lumped HBV model code implemented by MATLAB that is available at www.gloh2o.org (access date: 12 January 2020). The streamflow data, with its prediction interval and the uncertainties of parameters for the ungauged catchments produced during the current study, are available in the repository at https://doi.org/10.5281/ZENODO.5806358 (access date: 27 December 2021). Detailed descriptions of the datasets are shown by Abraham et al. [82]. The physical properties data such as permeability and porosity were extracted from the global datasets prepared by Huscroft et al. [44]. The catchment properties data extracted for gauged and ungauged catchments are shown in the manuscript in Table 2 and Table S2, respectively.

**Acknowledgments:** The authors would like to thank the German Academic Exchange Service DAAD and Hawassa University for the grant of this Ph.D. scholarship. We are thankful for all the structural facilities provided by the chair of hydrological modeling and water resources at the University of Freiburg. The Ministry of Water Irrigation and Electricity of Ethiopia provides the streamflow data used in this study. The authors would like to acknowledge the work of Beck et al. [83] and Beck et al. [30] who provided the MSWEP precipitation and HBV parameters, respectively, for the global scale. We are also thankful for the provision of the global scale potential evapotranspiration (GLEAM) datasets by Martens et al. [43].

**Conflicts of Interest:** The authors declare that they have no known competing financial interests or personal relationships that could have appeared to influence the work reported in this paper.

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
