# Peer review of "Prediction at Ungauged Catchments through Parameter Optimization and Uncertainty Estimation to Quantify the Regional Water Balance of the Ethiopian Rift Valley Lake Basin"

_hydrology, doi:10.3390/hydrology9080150_

Round 1

Reviewer 1 Report

I suggest this manuscript be accepted with very few technical things to fix.

In Chapter 3.1, exponents in units of measurement should be written as exponents.

The scale in the legend does not match the values on the map in Figures 8d and 8f. For example in Figure 8f, on the map marked CV = 59.45%, which according to legend should be in the middle of the scale (hence yellow), is marked in red.

Also, in Figures 8a-d, the values on the maps exceed the values on the scales.

The quality of individual images needs to be refined, for example Figure 8.

Reviewer 2 Report

The authors have done a good amount of work to justify the evaluation of HBV model by considering the Uncertainty Estimation Framework for Quantifying the Regional Water Balance for the Data-Scarce Region of the Ethiopian Rift Valley Lake Basin. However, there are some points described below that have to be considered before publication. For instance, though authors have mentioned the literature survey part, it fails to provide clear view on the previous attempts for understanding the different hydrological models. The introduction section needs some rework and restructuring to make a precise outline of study. So the authors need to narrow down the manuscript. For example, the authors should provide a better discussion of the results, and present the reasons behind the findings of this study. Furthermore, the manuscript contains grammatical errors that should be corrected. The manuscript should be read and corrected by a native English speaker.

Specific comments:

** Title**

Please, have a think about the title and whether it will be attractive to a wide audience - emphasize the fundamental science aspects of the work.

**Abstract**

What do the authors mean by “RVLB”?

Line 18, sentence is long and hard to understand for readers. Rephrase the line the sentence is incorrectly formed.

Please mention model simulation results in abstract part.

**Introduction**

Introduction could be restructured to better establish the main objectives of the study. Please ensure the hypotheses are clearly given and testable and that they relate to the aims and the objectives of the work.

Authors are required to mention the limitations or assumptions considered in this study at the end of Introduction.

**Data**

Authors are requested to add equation used in the HBV model for generating the streamflow and other water balance components.

In addition, as suggested from the elevation difference how the elevation control precipitation is accounted in the model. Please justify by giving physical equation governing precipitation in the model at various elevation.

Please provide the reference if authors themselves have not developed the equations wherever they have mentioned.

Please define new findings of your work which was not done by others.

**Methods**

What is the unique innovation of this research? Which gaps of previous study does it cover?

Section of appropriate inputs variable is important in the modeling process. Therefore, authors should add more references in model data section to support the parameters they selected as inputs. Generally, we have to do some test to prove the necessity of each variable.

How authors did reliability of the regionalization approach?

How authors did the selection of parameters? Which technique used for selection of parameters? Please explain

How authors did calibration and validation in this study and why it is feasible, please provide the details?

How authors computed R2 and NSE?

**Results discussion**

What is the reason relate your results with hydrological processes and I suggest authors to add water balance components as there are several studies showing these components (e.g., Srivastava et al., 2017) using HBV and compare their results with the previous studies top showcase how other hydrological processes responds to this input variations. I would recommend authors to look the study mentioned and other also and strengthen their discussion accordingly.

I would suggest authors to provide a result showing the segregation of overall rainfall time series into low, medium and high rainfall time series (RF is rainfall, µ is mean of rainfall, and σ is standard deviation of rainfall).

**Conclusion**

Make sure the conclusions are supported by the data presented.

Please, make sure the references are up to date and that you have checked recent issues of Hydrology for relevant papers including papers in press.

Please summarize the conclusion part.

In conclusion, the manuscript needs minor revision. I hope the comments and suggestions above may be of helping to the authors for preparing a revised version of the manuscript. 

Reviewer 3 Report

In this article useful information on An Uncertainty Estimation Framework for Quantifying the Re- 2 gional Water Balance for the Data-Scarce Region of the Ethio- 3 pian Rift Valley Lake Basin has been provided. However, author need to address following comments in order to publish in this journal.

1.      Introduction is very general without any data. Author should add overall energy consumption and emission data . They can refer following most related articles for related information: https://doi.org/10.1016/j.desal.2017.03.009

2.      Figure 1 & 2 need more explanation.

3.      They should add more results and detailed explanation.

4.      They should provide more detain on results.

5.      They should add economic analysis and compare with conventional processes.

6.      Overall English need to improve.

Article need minor revision for publication.
